# MAGIC: NEAR-OPTIMAL DATA ATTRIBUTION

## ABSTRACT

The goal of data attribution is to estimate how adding or removing a given set of training datapoints will affect model predictions. This goal is straightforward in convex settings, but in non-convex settings, existing methods are far less successful. Current methods' estimates often only weakly correlate with the ground truth. In this work, we present a new data attribution method (MAGIC) that combines classical methods and recent advances in metadifferentiation (Engstrom et al., 2025) to nearly optimally estimate the effect of adding or removing training data on model predictions at the cost of only 2-3x the cost of training a single model. MAGIC essentially "solves" data attribution as it is currently studied, thus enabling downstream applications and motivating more fine-grained future evaluations.

## 1 INTRODUCTION

A common objective when building machine learning systems is to predict *counterfactuals* about model behavior. For example, scaling laws (Kaplan et al., 2020; Muennighoff et al., 2023) aim to predict the performance of systems trained with more data and compute than is currently available; interpretability techniques (Kim et al., 2018) predict how models behave under counterfactual inputs.

In this work, we study *predictive data attribution*, or *datamodeling* (Ilyas et al., 2022), where the goal is to predict how a model would behave if it trained on a different dataset. This well-studied problem encompasses, e.g., estimating the effect (on the resulting model's predictions) of modifying a training example (Koh & Liang, 2017), removing a group of training examples (Koh et al., 2019; Bae et al., 2022; Park et al., 2023), or adding new training data sources (Ley et al., 2024).

Predictive data attribution in large-scale settings is challenging: it requires simulating training a model on a different dataset without actually training (Guu et al., 2023; Ilyas et al., 2024). In "classical" settings—when learning corresponds to minimizing a convex loss—statistical tools like the influence function (Hampel, 1947) accurately and efficiently estimate how different training data choices change model predictions (Rad & Maleki, 2018; Koh et al., 2019; Giordano et al., 2019). However, in the non-convex settings that are ubiquitous in natural domains like language/vision, current methods are less effective. Indeed, the best existing methods produce estimates that typically (a) only *moderately correlate* with the ground truth (Basu et al., 2021; Bae et al., 2022; Park et al., 2023) and (b) incur large absolute error (Bae et al., 2022).

### 1.1 CONTRIBUTIONS AND ROADMAP

What is the underlying cause of the gap between the convex and non-convex settings? One intuitively appealing explanation is that in contrast to training a linear or logistic regression model, training a deep neural networks is a complex and uninterpretable process. Perhaps data attribution is one of many things that is easy in the convex setting and difficult in modern machine learning. Against this backdrop, the main contribution of our work is best stated as:

> *Contrary to this intuition, we present a method that provides near-perfect predictions of model behavior as a function of the training data.*

Our method, called MAGIC, thus makes substantial progress towards *solving* the problem of data attribution (at least, as it is currently defined). The roadmap for the rest of the paper is as follows:

**Single-model data attribution.** Our point of start is the observation that the inherent randomness of large-scale training makes attributing specific model predictions to training data conceptually

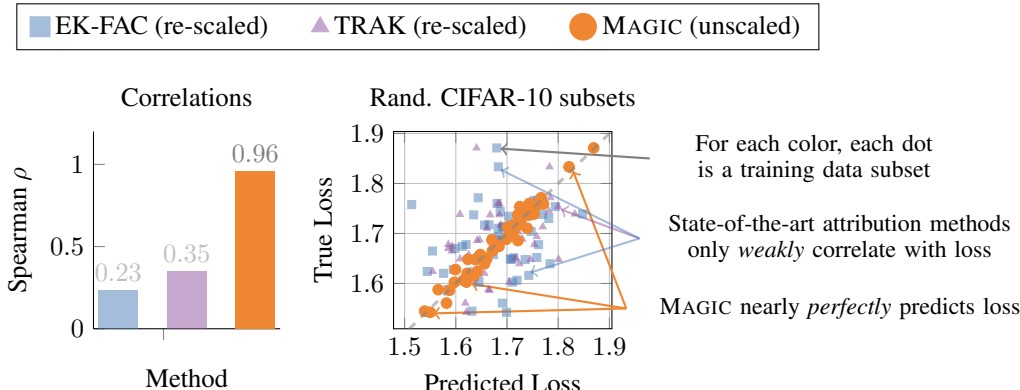

Figure 1: MAGIC nearly perfectly predicts the effect of training data removal. In contrast to the closest baselines (Park et al., 2023; Grosse et al., 2023), MAGIC estimates (a) more highly correlate with the ground truth effect and (b) are well-scaled. **Right:** we plot the predicted loss (of MAGIC and the baselines) against the true loss for a randomly chosen test point, each point a training data subset with 1% of samples randomly removed. For MAGIC, we plot the predicted loss since it is well-scaled; for the baselines, we first rescale the predictions to match the variance of the ground-truth losses. **Left:** The average (taken across test examples) Spearman correlation between predicted and true model losses (also known as the LDS (Ilyas et al., 2022; Park et al., 2023), see Section 2.1).

challenging (as also observed by prior work (Bae et al., 2022; Ilyas et al., 2022; Nguyen et al., 2023)). After all, if training on the same data can lead to different models, then we *cannot* predict the variation between these models as a function of the dataset. As a result, most prior data attribution methods can only predict how a *learning algorithm* would behave (*on average*) if trained on different data, but not how a *specific* model would behave if the training data changed.

Motivated by this state of affairs, we study a setting called "single-model" data attribution. The goal in this setting is still to predict the behavior of a model under changes to the training data—the twist is that we aim to predict how *a specific model* would have behaved under different training data, rather than how a newly initialized and trained model would have behaved under different training data. Originally proposed to study the temporal dynamics of training (see, e.g., (Wang et al., 2025)), we focus on the single-model setting for two reasons, which we make rigorous in Section 2. First, in the single-model setting, it *is* possible to perfectly predict model behavior as a function of the training data. Second, the predictions made in this setting correspond to how a given "single model" would respond to changes in the training data (motivating the name of the setting), rather than a given learning algorithm.

**A new data attribution method (MAGIC).** We present MAGIC (**M**etagradient-based **A**ttribution via **G**round-truth **I**nfluence **C**omputation), a state-of-the-art data attribution method. Our method leverages recent advances in large-scale metagradient calculation (Engstrom et al., 2025) to *exactly* calculate the influence function for large-scale learning algorithms. MAGIC accurately estimates how model predictions respond to random training data deletions—substantially outperforming existing methods—*even in the more challenging single-model setting*. For example (see Figure 1),

- When dropping different random 1% subsets from the training set of a ResNet-9 model trained on CIFAR-10, MAGIC almost exactly predicts ground-truth model losses ($\rho = 0.96$) while existing methods' predictions are only weakly correlated ($\rho = 0.25$).
- When dropping different random 1% subsets from the training set of a Gemma-2B model trained on instruction tuning data, MAGIC nearly exactly predicts ground-truth model test losses ($\rho = 0.97$) while existing methods perform no better than random guessing.

Our method is *optimal* for the family of evaluation metrics we consider, in that there is (provably) no way to improve performance along one metric without sacrificing another.

## 2   DATA ATTRIBUTION: NOTATION AND PROBLEM SETUP

The high-level goal of data attribution is to connect the choice of training data to model behavior. For example, one may want to use data attribution to find the training datapoints that cause a given output, or to surface data that harms accuracy. In this section, we formalize this goal with the *predictive data attribution* (or *datamodeling* (Ilyas et al., 2022)) framework, which phrases data attribution as the task of predicting how model behavior changes as a function of the training data.

Specifically, we view the machine learning pipeline as a three-step process wherein we (a) choose training data; (b) apply a learning algorithm to that data, yielding a trained model; and then (c) evaluate the trained model. The goal of predictive data attribution is to construct a function that *directly* predicts the output of step (c) from the choice of training data in step (a). To make this more precise (and borrowing from (Ilyas et al., 2022)), we define the following notation:

- Let $S = \{z_i\}_{i=1}^n$ be a pool of $n$ possible training examples. We represent *datasets* as vectors $\mathbf{w} \in \mathbb{R}^n$ where each entry $w_i$ is an importance weight for the $i$-th example in $S$. The importance weight $w_i$ controls the scaling of the loss of sample $i$; for example, $w_i = 0$ implies that we do not include the $i$-th example in the training set, while $w_i > 0$ implies that we do include the example (and multiply its loss by $w_i$ during training).
- Let $\mathcal{A} : \mathbb{R}^n \to \Theta$ be a *learning algorithm* mapping datasets—parameterized by importance weight vectors $\mathbf{w}$—to trained model parameters. We assume that all aspects of the training setup beyond the training data are captured by $\mathcal{A}$ (e.g., learning rate, weight decay, etc.).
- Let $\phi : \Theta \to \mathbb{R}$ be a *measurement function* mapping a machine learning model $\theta$ to a scalar measurement $\phi(\theta) \in \mathbb{R}$. For example, $\phi(\theta)$ might represent the loss of the classifier with parameters $\theta$ on a given test sample.
- Let $f : \mathbb{R}^n \to \mathbb{R}$ be the *model output function $f$* mapping datasets directly to model outputs, i.e., a composition of $\mathcal{A}$ and $\phi$.

To illustrate this notation, we instantiate it in the context of linear regression. In this case, the training pool is a set of $n$ input-label pairs $S = \{(\mathbf{x}_i \in \mathbb{R}^d, y_i \in \mathbb{R})\}_{i=1}^n$; the learning algorithm $\mathcal{A}$ fits a linear model minimizing the average squared loss *weighted* by a given $\mathbf{w} \in \mathbb{R}^n$, and the measurement function $\phi$ evaluates a model's loss on a specific test point $\mathbf{x}_{test}$, i.e.,

$$\mathcal{A}(\mathbf{w}) := \arg\min_\theta \sum_i w_i \cdot (\theta^\top \mathbf{x}_i - y_i)^2 \qquad \text{and} \qquad \phi(\theta) := (\theta^\top \mathbf{x}_{test} - y_{test})^2.$$

Then, $f(\mathbf{w}) := \phi \circ \mathcal{A}$ maps a data weighting $\mathbf{w}$ to the resulting model's loss on $\mathbf{x}_{test}$. Now, in this linear regression example, $f(\mathbf{w})$ is easy to directly compute (in fact, it has a closed form in terms of $\mathbf{w}$), but this is seldom the case. In general, evaluating $f(\mathbf{w})$ requires re-training a model on the weighting vector $\mathbf{w}$—which can make be very expensive for large-scale models. This motivates our (informal) definition of predictive data attribution, given in Definition 1 below.

---

**Definition 1** (Predictive data attribution). *A predictive data attribution is an explicit function $\hat{f}$ that approximates a model output function $f$. That is, for a given data weight vector $\mathbf{w}$, $\hat{f}(\mathbf{w})$ should be fast to compute while also accurately predicting $f$, i.e., satisfying $\hat{f}(\mathbf{w}) \approx f(\mathbf{w})$.*

---

### 2.1   SINGLE-MODEL PREDICTIVE DATA ATTRIBUTION

Our main goal is to operationalize predictive data attribution for large-scale (deep) learning algorithms. A challenge, however, is that these learning algorithms are *non-deterministic*, meaning that the same training dataset can map to *different* model parameters depending on randomness (e.g., random parameter initialization or data order shuffling) (Zhuang et al., 2022; Jordan, 2024b).

As a result, when learning algorithms are non-deterministic, we cannot perfectly predict how choice of data will change model behavior; we can only predict how it will change the *expected* behavior. More precisely, data attribution methods predict how (on average, over training randomness) a *new* model would behave if retrained from scratch, *and not how a specific trained model would behave if we had changed the training dataset*.

To make this more precise, let $\mathbf{w}$ be a weighting vector defining a dataset, and let $\hat{f}$ be a data attribution method. Now, consider the expected difference between our estimator $\hat{f}(\mathbf{w})$, and the true

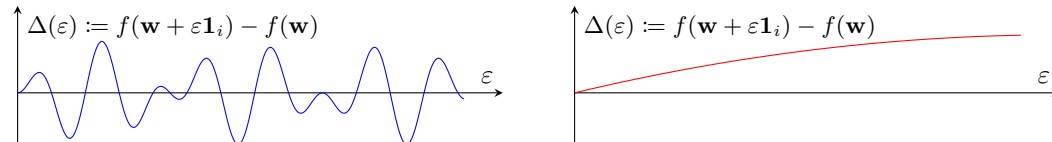

Figure 2: Smoothness aids predictive data attribution. We plot the change in data weights $\varepsilon$ against the change in model output $\Delta(\varepsilon)$ for two hypothetical learning algorithms. On the left is a non-smooth setting, where no simple function can easily approximate $\Delta(\varepsilon)$ (and thus no simple function can approximate $f(\mathbf{w})$). On the right is a smooth setting where the change is well-behaved.

model output $f(\mathbf{w})$, averaged over training randomness. This error decomposes into two terms:

$$\mathbb{E}\left[(\hat{f}(\mathbf{w}) - f(\mathbf{w}))^2\right] = \underbrace{\left(\hat{f}(\mathbf{w}) - \mathbb{E}[f(\mathbf{w})]\right)^2}_{\text{Reducible Error}} + \underbrace{\mathbb{E}\left[(f(\mathbf{w}) - \mathbb{E}[f(\mathbf{w})])^2\right]}_{\text{Irreducible Error}}. \tag{1}$$

Looking at each term in (1): the *reducible error* (or bias) is minimal when $\hat{f}(\mathbf{w}) = \mathbb{E}[f(\mathbf{w})]$, while the irreducible error (or variance) depends only on $f$, and is constant regardless of the data attribution method $\hat{f}$. Indeed, the irreducible error arises from inherent randomness in the model training process, and thus is fundamentally *unattributable* to data. Accordingly, current data attribution methods can answer questions about algorithms (e.g., *"what would happen if we trained a new model on a dataset not containing the training example $x$?"*)—but not about individual models.

However, in practice we often want to answer questions about *individual* models, not a class of learning algorithms. For example, we might ask a question like *"what was the effect of training example $x$ on this specific model?"* This question motivates us to define and consider a problem that we call *single-model data attribution*.

**Single-model predictive data attribution.** To understand how choice of data changes *individual* trained models, we study a setting called *single-model data attribution*. Here, we enforce that the learning algorithm $\mathcal{A}$ and the measurement function $\phi$ are deterministic (i.e., by fixing data ordering, parameter initialization, etc.). This determinism ensures that for any weighting $\mathbf{w}$, the model output $\phi(\mathcal{A}(\mathbf{w}))$ is deterministic (i.e., so that the expected model output that datamodels predict is constant, and $\text{Var}\left[\phi(\mathcal{A}(\mathbf{w}))\right] = 0$). In this new setting, model outputs vary only from changes in training data weights, allowing for predictive data attribution methods to exactly attribute changes to data weights. In the language of (1), the irreducible error is zero.

**Remark 1** (Single-model versus standard predictive data attribution). *Even when we care more about the average behavior of a learning algorithm and not a specific trained model, a near-perfect single-model data attribution method can be used to construct a near-perfect standard data attribution method by averaging over random seeds. We discuss this connection further in Section 6.*

## 2.2 SMOOTHNESS: A PREREQUISITE FOR DATA ATTRIBUTION

For predictive data attribution to even be possible (recalling Definition 1), the learning algorithm $\mathcal{A}$ must be, in some sense, well-behaved as a function of the data weights $\mathbf{w}$. Indeed, when this is not the case, it is unlikely that *any* simple predictor $\hat{f}$ will be able to accurately predict $f(\mathbf{w})$ from $\mathbf{w}$. Formally, the learning algorithm $\mathcal{A}$ should be *smooth* in $\mathbf{w}$ (Bubeck, 2014).

Although not usually made explicit, smoothness is an implicit assumption behind *any* data attribution method: to see why, take the following simple example (visualized in Figure 2). For a model output function $f$, consider slightly upweighting the $i$-th training sample, and measuring the change in the model output $\Delta(\varepsilon) := f(\mathbf{w} + \varepsilon \mathbf{1}_i) - f(\mathbf{w})$. If the learning algorithm is smooth, then this change is well-behaved as a function of $\varepsilon$. In particular, $\Delta(2\varepsilon)$ should be reasonably approximated by $2\Delta(\varepsilon)$. On the other hand, if the learning algorithm is *not* smooth, $\Delta(\varepsilon)$ may change wildly as $\varepsilon$ varies, precluding any simple predictor from being able to accurately predict $f(\mathbf{w})$ from $\mathbf{w}$.

**Remark 2** (How restrictive is smoothness?). *Smoothness is not an inherent property of standard learning algorithms, especially in the deterministic setting. However, it is necessary: for models resulting from non-smooth algorithms, there is no good data attribution method $\hat{f}$ that satisfies a*

*natural "additivity" property (Saunshi et al., 2023) (ruling out essentially all known data attribution methods). Fortunately, as observed by prior work, one can often construct a "smooth counterpart" to any given non-smooth learning algorithm (Engstrom et al., 2025).*

## 3  MAGIC: CALCULATING THE EXACT INFLUENCE FUNCTION AT SCALE

We now present MAGIC, our method for nearly-optimal data attribution. Our point of start will be a classical statistical technique called the influence function (Section 3.1). The influence function powers most large-scale data attribution methods, but, as we discuss in Section 3.2, the resounding success of the influence function in classical statistical settings does not seem to translate to modern ML setups. In Section 3.3, we introduce our method, MAGIC, which adapts the influence function to the modern ML setting by way of explicit metadifferentiation.

### 3.1  MOTIVATION: THE INFLUENCE FUNCTION

At the core of our method is a statistical primitive known as the *influence function approximation* (Hampel, 1947; Koh & Liang, 2017; Giordano et al., 2019). The main idea is to approximate the model output $f$ for a given data weighting $\mathbf{w}$ using the following first-order Taylor expansion:

$$\hat{f}(\mathbf{w}) := f(\mathbf{1}_n) + \left(\partial f(\mathbf{w})/\partial \mathbf{w}|_{\mathbf{w}=\mathbf{1}_n}\right)^\top (\mathbf{w} - \mathbf{1}_n). \tag{2}$$

The key term in this estimate is the gradient $\partial f(\mathbf{w})/\partial \mathbf{w}$ evaluated at $\mathbf{w} = \mathbf{1}_n$, called the *influence function*. Intuitively, this term captures the effect of infinitesimally up- or down-weighting each training example on the model output.

When the learning algorithm $\mathcal{A}$ is a *convex* optimization algorithm (e.g., linear regression, logistic regression), the influence function is straightforward to compute. Indeed, in such settings, the gradient $\partial f(\mathbf{w})/\partial \mathbf{w}$ has a simple closed form (via implicit differentiation), and the first-order Taylor expansion (2) yields near-perfect estimates of how the model output $f(\mathbf{w})$ will behave on new data weightings $\mathbf{w}$ (Koh & Liang, 2017; Rad & Maleki, 2018; Koh et al., 2019; Giordano et al., 2019).

### 3.2  THE INFLUENCE FUNCTION AND DEEP LEARNING

In large-scale, non-convex settings (e.g., in deep learning), no convenient closed form for the influence function exists. Instead, modern data attribution methods tend to approximate the influence function (Koh & Liang, 2017; Bae et al., 2022; Park et al., 2023; Bae et al., 2024). These methods have shown promise, but are not nearly as effective as the influence function is in convex settings. In fact, a thorough study conducted by Bae et al. (2022) reveals that for a variety of reasons (e.g., warm starting, early stopping, etc.), the influence function as instantiated by data attribution methods does not *even theoretically* predict the counterfactual effect of dropping data.

Given this landscape, we posit two possible explanations for the underwhelming performance of large-scale data attribution methods:

(a) The complexity of large-scale model training makes it hard to predict—and thus the influence function and all related methods are fundamentally limited.

(b) Existing methods do not do a good enough job at estimating the influence function.

To decide between these hypotheses, in the sequel we describe a technique for *exactly* computing the influence function for large-scale, non-convex models. In Section 4, we find that the resulting data attribution method (MAGIC) is extremely effective, both ruling out hypothesis (a) and making substantial progress on the data attribution problem as a whole.

### 3.3  CALCULATING THE EXACT INFLUENCE FUNCTION

We now discuss how to compute the exact influence function in the context of deep learning models.

**Iterative learning algorithms.** Recall from Section 2 that the learning algorithm $\mathcal{A}$ takes as input a data weighting $\mathbf{w}$ over the training set, and outputs a machine learning model trained on the weighted dataset. The algorithm thus encapsulates all aspects of the training setup beyond the training data

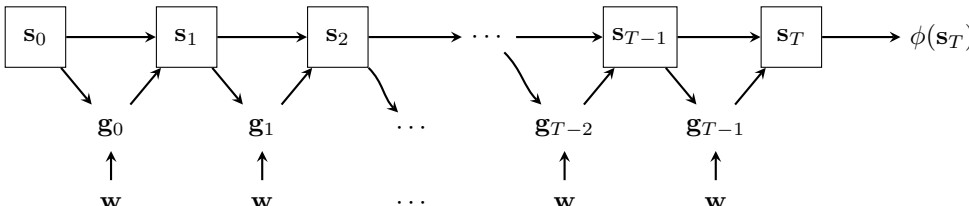

Figure 3: Forward computation graph for a model output function $f$ mapping from data weights $\mathbf{w}$ to the model output. The exact influence function $\partial f(\mathbf{w})/\partial\mathbf{w}$ is the *metagradient* of the model output with respect to the data weights $\mathbf{w}$.

weights, including the model architecture, optimizer, and hyperparameters. We observe that nearly all large-scale learning algorithms $\mathcal{A}$ are *iterative*, i.e., take the form

$$\mathcal{A}(\mathbf{w}) := \mathbf{s}_T \quad \text{for} \quad \mathbf{s}_{t+1} := h_t(\mathbf{s}_t, \mathbf{g}_t(\mathbf{s}_t, \mathbf{w})) \quad \text{and} \quad \mathbf{g}_t(\mathbf{s}_t, \mathbf{w}) := \sum_{i \in B_t} w_i \cdot \nabla_{\mathbf{s}_t} \ell(z_i; \mathbf{s}_t). \quad (3)$$

Above, $\mathbf{s}_t$ is the optimizer state (including model parameters), which is updated by a function $h_t$ starting from an initial state $\mathbf{s}_0$. The number of training steps is $T$; $B_t \subset [N]$ is a minibatch sampled at step $t$; and $\ell(z_i; \mathbf{s}_t)$ is the loss on sample $z_i$ given optimizer state $\mathbf{s}_t$. Stateful optimizers can be handled using the optimizer state $\mathbf{s}_t$, as we show in the following example.

**Example 1** (Training an LLM with Adam). *Here, the optimizer state $\mathbf{s}_t = (\theta_t, m_t, v_t)$, where $\theta_t$ is the parameter vector at step $t$, $m_t$ is the first moment estimate of the gradient, and $v_t$ is the second moment estimate of the gradient. The loss function $\ell(z_i; \mathbf{s}_t)$ is the loss of a language model with parameters $\theta_t$ on a given training example $z_i$, and the update function $h_t$ is the Adam update step:*

$$h(\mathbf{s}_t, \mathbf{g}_t) := \left[ \begin{array}{c} \theta_t - \eta_t \cdot \frac{\sqrt{v_t}}{\sqrt{m_t + \epsilon_{\text{root}}} + \epsilon} \cdot \mathbf{g}_t \\ \beta_1 \cdot m_t + (1 - \beta_1) \cdot \mathbf{g}_t \\ \beta_2 \cdot v_t + (1 - \beta_2) \cdot \mathbf{g}_t^2 \end{array} \right],$$

*where $\eta_t$ is the learning rate at step $t$.*

**Calculating the influence function.** To compute the exact influence function for a model output function $f$ that is the output of an iterative learning algorithm $\mathcal{A}$, we leverage recent developments in *metagradient* calculation (Maclaurin et al., 2015; Franceschi et al., 2017; Engstrom et al., 2025). A metagradient is a gradient of a machine learning model's output with respect to a design choice made prior to training. (For example, imagine taking the derivative of a model's test loss with respect to the learning rate used to train the model.) In recent work, Engstrom et al. (2025) give an algorithm called REPLAY that *exactly* calculates the metagradient for iterative and smooth learning algorithms.

Observe that when the design choice of interest is the data weighting $\mathbf{w}$, the metagradient—the gradient of the model output $f$ with respect to $\mathbf{w}$—is precisely the influence function. We can thus directly apply REPLAY (Engstrom et al., 2025) to calculate the exact influence function. Adapted to our setting, REPLAY calculates the metagradient by exploiting the following identity, which follows from the chain rule applied to the computation graph in Fig. 3:

$$\nabla_{\mathbf{w}} f(\mathbf{w}) = \sum_{t=0}^{T-1} \underbrace{\frac{\partial f(\mathbf{w})}{\partial \mathbf{s}_{t+1}} \cdot \frac{\partial h_t(\mathbf{s}_t, \mathbf{g}_t(\mathbf{s}_t, \mathbf{w}))}{\partial \mathbf{w}}}_{\text{contribution of } \mathbf{w} \text{ to } f(\mathbf{w}) \text{ through } \mathbf{s}_{t+1}}. \quad (4)$$

Motivated by this observation, REPLAY operates by initializing $\Delta_T = \frac{\partial f(\mathbf{w})}{\partial \mathbf{s}_T} = \nabla_{\mathbf{s}_T} \phi(\mathbf{s}_T)$, then

1. For each $t = T-1, \ldots, 0$:
   (a) Load state $\mathbf{s}_t$ and minibatch $B_t$
   (b) Calculate $\beta_t = \nabla_{\mathbf{w}} \left( h_t(\mathbf{s}_t, \mathbf{g}_t(\mathbf{s}_t, \mathbf{w}))^\top \Delta_{t+1} \right)$, the contribution of $\mathbf{w}$ to $\phi(\mathbf{s}_T)$ through the $t$-th step of the learning algorithm
   (c) Advance $\Delta_t = \nabla_{\mathbf{s}_t} \left( h_t(\mathbf{s}_t, \mathbf{g}_t(\mathbf{s}_t, \mathbf{w}))^\top \Delta_{t+1} \right)$, which is $\partial f(\mathbf{w})/\partial\mathbf{s}_t$
2. Return $\beta = \sum_{t=0}^{T-1} \beta_t$ as the exact influence function

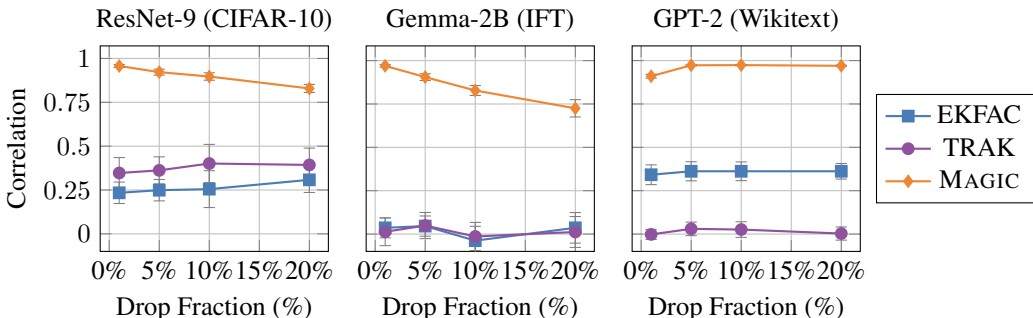

Figure 4: Linear datamodeling score (LDS) vs. drop fraction across settings for MAGIC and baselines. The estimates of MAGIC consistently correlate with the true model outputs (LDS: near $1.0$ for small enough drop fraction) while baselines often do not (LDS: below $0.4$). LDS decreases with increasing drop fraction for MAGIC (as the Taylor estimate moves further from the center).

By leveraging an efficient data structure to load the states and minibatches, REPLAY is able to calculate the exact influence function at a computational cost of $T + T \log(T)$ total training steps and $\log(T)$ memory. We refer the reader to Engstrom et al. (2025) for a complete description.

## 4 EVALUATION

In this section we evaluate MAGIC across a number of domains. We compare MAGIC with two of the most successful recent data attribution techniques: TRAK (Park et al., 2023) and EK-FAC (Grosse et al., 2023); see Appendix B for the specifics of these baselines. Across the board, we find that MAGIC provides near-perfect predictions of how model outputs change when we drop data (at random) from the training set.

**Evaluation metric.** Recall (from Section 2) that the goal of predictive data attribution is to predict how a model's output changes as a function of the model's training data. In order to evaluate the quality of these predictions, we adopt the *linear datamodeling score* (LDS) (Ilyas et al., 2022; Park et al., 2023) as our evaluation metric. To compute LDS for a given model output function $f$ and corresponding data attribution method $\hat{f}$, we follow the following steps:

1. Sample $n$ fixed-sized subsets of the training set, which we represent as binary data weights $\mathbf{w}^{(1)}, \ldots, \mathbf{w}^{(n)} \in \{0,1\}^N$, where $N$ is the number of total training samples. Given a drop-out fraction $p \in [0,1]$, we sample each vector $\mathbf{w}^{(i)}$ by dropping $pN$ random training samples.
2. For each data weight vector $\mathbf{w}_i$, we (a) compute the *true* model output $f(\mathbf{w}_i)$ by training a model on the train set with data weights $\mathbf{w}_i$ and evaluating the measurement of interest on the trained model, and (b) compute the *predicted* model output $\hat{f}(\mathbf{w}_i)$ via the data attribution method $\hat{f}$.
3. We compute the LDS as the Spearman correlation between the predicted output and the true output over all $n$ data weight vectors, i.e.,

$$\text{LDS} = \rho\Big( \Big[\hat{f}(\mathbf{w}_i)\Big]_{i=1}^n, \Big[f(\mathbf{w}_i)\Big]_{i=1}^n \Big). \tag{5}$$

**Settings.** We study scenarios spanning computer vision and language modeling. Each scenario comprises a training dataset $S$, a learning algorithm $\mathcal{A}$, and a test set $S_T$. Accordingly, each scenario defines $|S_T|$ data attribution tasks, where task $i$ is to predict the loss of $\mathcal{A}$ on the $i$-th sample in $S_T$. For each data attribution method we consider, we compute the *average* LDS (5) across tasks.

- **ResNet CIFAR-10 training**: We train ResNet-9 (Jordan, 2024a) models on subsets of the train set of CIFAR-10 (Krizhevsky, 2009) and aim to predict cross-entropy loss on its test samples.
- **GPT-2 Wikitext fine-tuning**: We fine-tune 125M GPT-2 models on subsets of Wikitext and aim to predict (language modeling) loss on 50 different test samples.
- **Gemma-2B instruction-tuning**: We fine-tune Gemma-2B with LoRA on subsets of 3 combined instruction tuning datasets (Flan V2, DOLLY, OA-1), aiming to predict loss on MMLU samples.

See Appendix A for the exact details of each scenario.

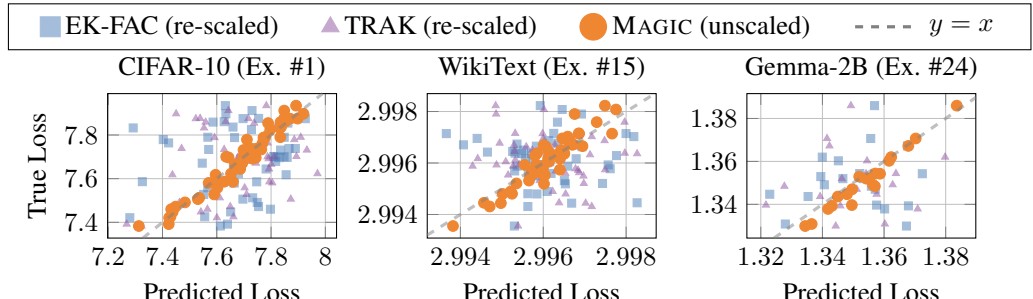

Figure 5: Results of MAGIC and baselines on randomly chosen, individual samples from three settings: CIFAR-10, Gemma-2B, and GPT-2. We evaluate by predicting model output after dropping a random 5% of the data (cf. (5)) and plotting the results against the true model output for that drop set. MAGIC estimates consistently highly correlate with the true output across settings.

## 4.1 RESULTS

As shown in Figure 4, MAGIC attains near-perfect LDS across settings and drop-out fractions, although our predictions slightly degrade as we drop more data. Existing baselines are noisy in comparison; these methods' predictions only weakly correlate with the ground truth model losses.

In Figure 5, we randomly select a test example from each scenario, and plot predictions of test loss against true test loss for each data attribution method. MAGIC almost exactly predicts the true test loss, even in absolute terms. On the other hand, baselines barely correlate with the true predictions, and are mis-scaled in absolute terms (TRAK and EKFAC predictions are not of the right order of magnitude, so we rescale them to visualize them on the same plot).

**Optimality of MAGIC.** We observe that the performance of MAGIC degrades with the fraction of samples dropped. While MAGIC has near-perfect LDS when predicting the effect of removing a small fraction of the training data (i.e., 1%), the LDS degrades at larger drop-out fractions (i.e., 20%). Our near-perfect performance when dropping only a few points indicates that our method is in some sense optimal among linear predictors: we can perfectly predict in a small ball around "not dropping out any points," but curvature (in training data weight space) causes the linear approximation to degrade further away from training on all the data. In other words, there is no way to significantly improve the performance upon of MAGIC at larger drop fractions without simultaneously hurting its performance elsewhere.

**Other baselines.** In this section, we focus on comparing MAGIC to TRAK and EK-FAC, which are two highly successful data attribution methods with reasonable runtime and open-source code available. In some sense, however, the choice of baseline does not matter—as far as we are aware, no existing data attribution method attains near-perfect performance across settings and benchmarks. To further put this result into context:

- Wang et al. (2025), who also study the single-model data attribution setting, report an LDS of 0.715 for a two-layer CNN trained on the MNIST dataset. In the exact same experimental setting, MAGIC achieves an LDS of 0.998.
- Bae et al. (2024), who use a single model to compute data attribution but still evaluate in the standard "average-model" setting, report a (state-of-the-art) LDS of 0.1 on CIFAR-10 when removing random 10% fractions of data. In the same experimental setup, MAGIC achieves an LDS of 0.6 (which is near-optimal, in that the upper bound, i.e., ground-truth correlation between a single model and the average, is 0.65).

## 5 RELATED WORK

Our work adds to the growing literature on (predictive) data attribution methods—see (Ilyas et al., 2024) or (Hammoudeh & Lowd, 2022) for a survey. Related to our method are those that use variants of the influence function (Hampel et al., 2011), i.e., the Taylor approximation from Section 3.1. In settings where the learning algorithm minimizes a convex objective, such influence function-based

methods are known to have an efficient and accurate closed form (Rad & Maleki, 2018; Koh et al., 2019; Giordano et al., 2019; Nobel et al., 2024). In scenarios where the learning algorithm does not return a convex minimizer (such as in deep learning), this closed form is not available. In such cases, the dominant approach is to apply one of many efficient approximate approaches (Koh & Liang, 2017; Ladhak et al., 2022; Schioppa et al., 2022; Park et al., 2023; Grosse et al., 2023; Wang et al., 2024). However, in the non-convex setting, these approximations do not offer correctness guarantees like they do in convex settings, and can even have different interpretations entirely (Bae et al., 2022)—potentially leading to the poor correlations we observe in Section 4. Close to our work are methods that aim to approximate the influence function via *unrolling* (Bae et al., 2024). These methods leverage the same recursive formula of (4)—but still *approximate* the influence function rather than compute it exactly.

To compute the influence function exactly, we leverage recent advances in *metagradient* calculation (Engstrom et al., 2025), which build on a long line of work on differentiating through optimization (Maclaurin et al., 2015; Lorraine et al., 2020). See Engstrom et al. (2025) for a recent survey.

Finally, our single-model data attribution setting is motivated by the nondeterminism of model training. This phenomenon has been studied from a variety of perspectives, including training dynamics (Zhuang et al., 2022; Jordan, 2024b), fairness (Black et al., 2022; Marx et al., 2020), and even data attribution (Ilyas et al., 2022; Nguyen et al., 2023).

## 6 DISCUSSION AND CONCLUSION

In this section, we discuss the connections between our single-model data attribution setting and the standard data attribution problem, the computational cost of our MAGIC compared to the baselines we consider, along the way mentioning limitations of our method.

**Single-model versus standard data attribution.** From a conceptual perspective, while standard data attribution is about predicting how a *new* model would behave if retrained from scratch on a counterfactual dataset, single-model data attribution is about predicting how a *specific* model would behave if the data had been different. Technically, a *perfect* single-model data attribution method can exactly predict the behavior of a specific model on any counterfactual dataset. By averaging these predictions over multiple models (each corresponding to a different learning algorithm seed), we obtain a perfect "standard" predictive data attribution method. The other direction, however, does not hold: it may be possible to construct a predictive data attribution method that perfectly predicts the average behavior of a learning algorithm, but not the behavior of a specific model.

**Computational cost of data attribution.** One consideration that we have not yet addressed is the computational cost of *building* the data attribution method $\hat{f}$. For a single model output function $f$, the cost of building $\hat{f}$ with MAGIC is only a small constant factor (e.g., 2-3x) more than training a model itself, 1-2 orders of magnitude faster than the baselines. The costs differ, however, in how they scale with the number of output functions considered. For example, consider a setting with $N$ training samples and $n$ distinct output functions. Methods like TRAK and EK-FAC have runtime $C \cdot (N + n)$ for some very large constant $C$, while the runtime of MAGIC is roughly $c \cdot N \cdot n$ for a constant $c$ typically 10-100x smaller than $C$. As a result, MAGIC is *significantly* faster than TRAK and EK-FAC when $n$ is small—for example, when $n = 1$ (attributing a single output function) on the CIFAR-10 dataset, MAGIC costs essentially 2-3x as much as training a single model (15 minutes on a single A100 GPU), while TRAK and EK-FAC are both 20-1000x slower. As $n$ increases, however, MAGIC's cost increases linearly, while TRAK and EK-FAC's costs stay roughly constant.

Note that an output function does not necessarily correspond to a single test example: attributing a model's average performance across a group of test examples requires only a single output function, making MAGIC a useful tool for efficiently understanding the sources of, e.g., specific capabilities.

**Conclusion.** we present MAGIC, a new data attribution method that near-exactly predicts how model outputs change as a function of its training data (according to standard metrics). To do so, MAGIC operates by calculating the exact influence function using recent advances in metadifferentiation (Engstrom et al., 2025). Given the magnitude at which MAGIC improves our ability to estimate the effect of training data at high fidelity, we are excited to see what downstream applications for data attribution MAGIC unlocks, including near-perfect unlearning (Georgiev et al., 2024), model debugging (Koh & Liang, 2017; Shah et al., 2022), and more.

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

# A    EXPERIMENTAL DETAILS

In this section, we provide additional details on the experimental setup used in the main paper, including the training details of the models, and the datasets used.

**ResNet-9 on CIFAR-10.** We use the ResNet-9 architecture from (Jordan, 2024a), with the hyperparameters given in Table 1. To give concrete details: the training set $S$ comprises $50,000$ CIFAR-10 training samples, the learning algorithm $\mathcal{A}$ is standard supervised training, and we consider 50 measurement functions $\phi_i$ corresponding to loss on 50 different CIFAR-10 test samples.

| Hyperparameter | Value |
|---|---|
| Learning rate | 1.2 |
| Weight decay | 0.001 |
| Bias scale | 8.0 |
| Batch size | 1000 |
| Epochs | 12 |
| Final layer scale | 0.04 |
| Momentum | 0.875 |
| Pooling type | Log-sum-exp |
| Pooling $\varepsilon$ | 0.1 |
| Width multiplier | 2.5 |
| LR schedule | One-cycle Linear |
| LR start multiplier | 0.07 |
| LR end multiplier | 0.2 |
| LR peak time | 0.5 |

Table 1: Hyperparameters for ResNet-9 on CIFAR-10.

**Gemma-2B LoRA on IFT Data.** We use the variant of LESS (Xia et al., 2024) from Engstrom et al. (2025). In particular, the training dataset consists of the four instruction fine-tuning sets seen in Table 2 as in LESS. The total number of points is around $300,000$ and is exactly four combined IFT datasets (Flan V2 (Longpre et al., 2023), CoT (Wei et al., 2022), DOLLY (Conover et al., 2023), and Open Assistant 1 (Köpf et al., 2024)). We test on a randomly chosen (task balanced) subset of of MMLU comprising 32 test samples. We use 4-shot in-context learning for these samples. We adapt a LoRA to a Gemma-2B model (the pretraining-only Gemma-2B model (Team et al., 2024)) using the LoRA configuration from Xia et al. (2024). For model training, we use the same setup as Engstrom et al. (2025), except with $\epsilon_{\text{root}} = 10^{-6}$. In particular, we train with ADAM ($\beta_1 = 0.95$, $\beta_2 = 0.975$, decoupled weight decay as $10^{-5}$) and a one-cycle linear schedule starting at $10^{-6}$ of the maximum learning rate, reaching the peak over 25% of training, then ending at $0.1$ of the maximum learning rate ($0.0004$). We insert a positive $\epsilon_{\text{root}}$ into the inverse square root term in the ADAM update to prevent metagradient (and to a lesser extent update) blowup.

**GPT-2 fine-tuning on Wikitext.** We optimize a pre-trained GPT2 (Radford et al., 2019) model on Wikitext (Foundation, 2022) using causal language modeling. We split the Wikitext samples into size 512 context length chunks and into train and test splits, with 256 samples in the test split and 4608 samples in the train split. We attribute on the test split, and use 4 epochs of the train split during training. We use the same ADAM optimizer setup above except that we set $\epsilon_{\text{root}} = 10^{-8}$, max learning rate to $0.0008$, and do not anneal ADAM $\epsilon_{\text{root}}$.

Table 2: Details of IFT training datasets.

| Dataset | # Instance | Sourced from | Prompt Len. | Complet. Len. |
|---|---|---|---|---|
| FLAN V2 | 100,000 | NLP datasets and human-written instructions | 355.7 | 31.2 |
| CoT | 100,000 | NLP datasets and human-written CoTs | 266 | 53.2 |
| Dolly | 15,011 | Human-written from scratch | 118.1 | 91.3 |
| Open Ast. 1 | 55,668 | Human-written from scratch | 34.8 | 212.5 |

# B BASELINES

In this section, we provide a brief overview of the baselines we consider. The basis of both of these methods is similar, and rooted in the Taylor approximation (2) that also underpins MAGIC:

$$\hat{f}(\mathbf{w}) := f(\mathbf{1}_n) + \left( \frac{\partial f(\mathbf{w})}{\partial \mathbf{w}} \bigg|_{\mathbf{w}=\mathbf{1}_n} \right)^\top (\mathbf{w} - \mathbf{1}_n);$$

## B.1 TRAK (TRACING THE RANDOMLY-PROJECTED AFTER KERNEL)

TRAK (Park et al., 2023) estimates the influence of individual training examples on model predictions. However, instead of computing exact the influence, TRAK calculates the influence for a simple proxy model that (a) is easy to calculate the influence for and (b) is meant to match the original model class of interest. In particular, TRAK *approximates* the original learning algorithm, $\mathcal{A}$, by linearizing around the final parameters. We refer the reader to Park et al. (2023) for full details.

## B.2 EK-FAC (EIGENVALUE-CORRECTED KRONECKER-FACTORED APPROXIMATE CURVATURE)

EK-FAC (Bae et al., 2022; Grosse et al., 2023) is another influence-based data attribution method. To estimate the influence function, the method estimates the Hessian via the Fisher information matrix/Gauss-Newton hessian (Martens & Grosse, 2015; George et al., 2018), then applies a version of the infinitesimal jackknife (Giordano et al., 2019) to calculate the gradient with respect to data weights. For an excellent high level overview of this approach, see Appendix D.1 of Bae et al. (2024).

