# OpenReview forum: "MAGIC: Near-Optimal Data Attribution for Deep Learning"
_ICLR.cc/2026/Conference — Submitted to ICLR 2026_

### Official Review · Reviewer_HE5Y · 2025-10-30

**Soundness:** 2
**Presentation:** 3
**Contribution:** 3
**Rating:** 6
**Confidence:** 3

**Summary:**

In this work, the authors propose MAGIC, a method for near-perfect data attribution in large-scale, non-convex deep learning models. Traditional approaches such as TRAK and EK-FAC approximate the relation between training data and model behavior using linearized influence functions, but they perform poorly in non-convex settings. To address this, the authors reformulate the problem under the single-model predictive data attribution framework, assuming deterministic learning and evaluation. They show that for smooth, iterative training algorithms, the influence of each training example on the model output can be exactly expressed as a metagradient with respect to data weights, and compute it efficiently using the REPLAY algorithm, which replays the full training trajectory backward in time.

The authors then evaluate MAGIC on CIFAR-10 + ResNet-9, Wikitext + GPT-2, and Gemma-2B instruction tuning on IFT dataset, showing that it achieves amazing performance on the Linear Datamodeling Score (LDS), significantly outperforming TRAK and EK-FAC.

**Strengths:**

1. The paper clearly explains the motivations and research gap, provides a well-defined description of the MAGIC, and highlights its key contributions in an organized manner.

2. The theoretical discussion is great, and the experimental design is carefully carried out.

3. The presentation and comparison of experimental results in this paper are very clear. For example, in Section 4.1, Figures 4 and 5 provide a clear comparison with the baselines.

**Weaknesses:**

1. The LDS as an evaluation metric here is not very reasonable to me. Although LDS is widely accepted in the research community, it is fundamentally designed for a specific model training configuration. In other words, it is not an appropriate metric for evaluating single-model data attribution. Perhaps downstream task evaluations related to the individual model would be more suitable.

2. This is minor, but do the authors use ensemble averaging to stabilize and improve TRAK’s results? If not, do the authors believe the comparison between MAGIC and TRAK/EK-FAC, which are not single model attribution methods, is fully fair? It might be more appropriate to use other single-model attribution methods, which are mentioned in Section 4.1, as the main baselines for this comparison.

**Questions:**

1. Could you provide the configuration details for TRAK (e.g., projection dimensionality and number of ensembles) and EK-FAC?

2. While this paper provides an insightful computational cost analysis, could you also quantify the exact runtime and memory costs of MAGIC compared to TRAK and EK-FAC under the same hardware and dataset scale? In particular, how does the cost of MAGIC scale with the number of training samples 𝑁 and the number of output functions 𝑛 (mentioned from 465 - 475) relative to these baselines?

---

### Official Review · Reviewer_Kqc5 · 2025-10-30

**Soundness:** 3
**Presentation:** 2
**Contribution:** 2
**Rating:** 4
**Confidence:** 4

**Summary:**

The paper aims at resolving the limitation of current data attribution methods in non-convex settings and propose a new data attribution method (MAGIC) that leverage metadifferentiation to "solve" the data attribution. The paper also claims that the method is reasonably efficient with relatively 2-3X cost of training a single model and boost some downstream tasks.

**Strengths:**

- The paper pointed out two important issues in current (influence function based) data attribution methods: non-convex approximation and single-model data attribution (randomness in training process).
- Equation (4) and  Figure 3 serve good for the technical foundation of this paper.
- The improvement over LDS is clear and significant.

**Weaknesses:**

- The motivation, solution, and evaluation is not well connected
  - IF based methods assume the model to be convex and bring limitations and performance when the model is non-convex; while the task of single model data attribution (introduced in section 2.1 and section 6) is another problem (e.g., non-converging, training algorithm randomness, ...)
  - The solution is essentially aim at replay (through calculating the iterative effect of data weighting to the f) the whole training process with same random seed, data order, .... which is reasonable for single model data attribution, but a little bit hard to see how the process resolve the non-convex issue.
  - The evaluation metric LDS is used for evaluation, but no specific change is made to the LDS (like average (5) LDS is still used). It's hard to see why this is a good evaluation metric to single model data attribution.

- The computational cost analysis is vague.
  - It can be seen that the MAGIC's computational source is comparable to the training cost, but it is hard to see why
    - "constant c typically 10-100x smaller than C"
    - "while TRAK and EK-FAC are both 20-1000x slower."
  - No storage cost analysis

- Missing important analysis with some related work on single-model data attribution
  - TracIN (can bee seen as single model data attribution when it use dense checkpoints)
  - DVEmb and SOURCE (discussed in a short paragraph in "Other baselines", but these might be the main baselines the paper may want to compare)
  - SGD-influence (compare the essential difference between MAGIC and SGD-influence)

- The presentation could be improved
  - The introduction data attribution preliminary, single model data attribution, and influence function take too much spaces while the method (from line 300 - 323) is quite short and is not self-contain. Readers need to read a substantial part of another paper (metagradient paper) to understand this part of the paper.

**Questions:**

- Is there any specific change to LDS metric to make it suitable for single-model data attribution evaluation?
- Any idea (no need to carry out experiments) to reduce the significant cost of MAGIC?

---

### Official Review · Reviewer_M8SL · 2025-11-01

**Soundness:** 2
**Presentation:** 3
**Contribution:** 2
**Rating:** 2
**Confidence:** 5

**Summary:**

This paper proposes a new method for data attribution called Metagradient-based Attribution via Ground-truth Influence Computation (MAGIC). In essence, the proposed method calculates a form of data influence that is previously known as unrolling-based (Bae et al. 2024) or trajectory-specific influence function (Wang et al. 2025). The key difference is that the proposed method leverages a recent technique that trades off between the storage and computation costs in recovering the optimizer state at each step along the training process, and calculates the exact gradients at each step (as opposed to approximate ones) to calculate the unrolling influence.

Empirically, the authors demonstrate high correlation of the proposed method with retraining after removing a small subset of training data. In the experiments, the proposed method is mainly compared with baseline methods (EKFAC and TRAK) that do not take unrolling into account. There are two one-sentence comments regarding the comparison of the proposed method with Bae et al. 2024 and Wang et al. 2025 without details. The computational cost of the proposed method is notably high: it requires 2-3x full training cost for each model prediction, with an additional storage cost for O(log T) copies of the optimizer states for T training steps.

**Strengths:**

- The proposed method achieves high LDS scores.
- The experiments are conducted on both image and text settings.
- The paper is clearly written and easy to follow.

**Weaknesses:**

- The proposed method is estimating a known form of data influence that has been recently investigated by Bae et al. 2024 [1] or Wang et al. 2025 [2], which dates back to at least Hara et al. 2019 [3]. However, the current paper fails to properly acknowledge the existing literature.
  + The term "single-model data attribution" is a bit misleading. Given that the problem formulation in this paper is identical those in either Bae et al. 2024 [1] or Wang et al. 2025 [2], it would be better to just call it "unrolling-based data attribution" or "trajectory-specific data attribution" instead of creating a new terminology referring to the same thing. IMO both terms are better than "single-model data attribution" by emphasizing the specific training trajectory. "Single-model data attribution" could be potentially confusing as conventional influence function also just uses a single model to calculate.
- The proposed method is a straightforward application of the metagradient [4]. In fact, the metagradient used in the application 4.1 in Engstrom et al. 2025 [4] seems to be the same as the proposed method. Thus the technical novelty of this work is limited.
- The proposed method suffers from significant computation and storage costs compared to existing methods [1, 2]. In particular, 2-3x full training costs **per model prediction** makes it hardly practical for large-scale applications.
- Detailed comparison to the most relevant baseline methods [1, 2] is missing.
- Only evaluated with LDS. No experiments on downstream applications.
- Overall, the claim that *MAGIC essentially "solves" data attribution* is over-claiming.


**References**
- [1] Bae, J., Lin, W., Lorraine, J., & Grosse, R. Training data attribution via approximate unrolled differentiation. NeurIPS 2024.
- [2] Wang, J. T., Song, D., Zou, J., Mittal, P., & Jia, R. Capturing the Temporal Dependence of Training Data Influence. ICLR 2025.
- [3] Hara, S., Nitanda, A., & Maehara, T. Data cleansing for models trained with sgd. NeurIPS 2019.
- [4] Engstrom, L., Ilyas, A., Chen, B., Feldmann, A., Moses, W., & Madry, A. Optimizing ml training with metagradient descent. Arxiv 2025.

**Questions:**

- Can the authors provide a detailed comparison that, mathematically, how is the proposed attribution score different from the metagradient g in the Section 4.1 of Engstrom et al. 2025 (arXiv:2503.13751v1)?
- The calculation of the metagradient seems to involve second-order terms ($\frac{\partial g_t}{\partial s_t}$) that are as expensive as Hessians. Could the authors provide a more detailed complexity analysis of the algorithm in terms of the model parameter dimension?
- Is the method applicable to full-parameter fine-tuning on LLMs (as opposed to just LoRA fine-tuning)?

---

### Official Review · Reviewer_Ft5A · 2025-11-01

**Soundness:** 3
**Presentation:** 3
**Contribution:** 2
**Rating:** 4
**Confidence:** 4

**Summary:**

This work studies the single-model-specific data attribution algorithm, an important problem in the field of data attribution. Specifically, the authors leverage the chain rule through the model training and leverage the recent advances in the meta-gradient algorithm to facilitate the exact computation for the influence function.

**Strengths:**

1. The presentation and illustration are clear and clean.
2. The problem considered is a growing interest in the field, addressing an important research question.

**Weaknesses:**

1. The main technical advance lies in the new meta-gradient algorithm proposed in (Engstrom et al., 2025), rendering the current work as a straightforward application of it.
2. The empirical evaluation is incomplete due to the lack of "appropriate baselines," such as SGD-Influence (Hara et al., 2019), Unrolled Differentiation (Bae et al., 2024), and Data Value Embedding (Wang et al. 2025).
3. The discussion on the smooth assumption mentioned in Remark 2 should be elaborated more, as it is the methodology MAGIC is based. For instance, if the assumption is violated, how large an error will be incurred?

**Questions:**

1. (Weakness 1) Are there any additional technical advancements besides (Engstrom et al., 2025)? For instance, are there any other technical challenges when applying the proposed meta-gradient algorithm?
2. (Weakness 2) Can the author also provide the experimental results with the appropriate baselines mentioned, e.g., SGD-Influence, Unrolled Diff., and DVEmb?
3. (Weakness 3) Can the author provide a detailed discussion on how large an error might be incurred when the assumptions are violated?

---

### Meta-Review · Area_Chair_DVra · 2025-12-29

**Summary:**

This paper proposes a new data attribution method, MAGIC. Most reviewers lean toward rejection. The main concerns are that the method seems to be a fairly direct application of the recently proposed method by Engstrom et al. (2025), and that the comparison and discussion with closely related previous work, particularly Bae et al. (2024) and Wang et al. (2025), are not sufficiently thorough. Reviewers also note that the proposed method incurs relatively high computational cost. Unfortunately, these limitations and concerns were not addressed by the authors during the rebuttal period.

**Reviewer Concerns:**

No rebuttal was submitted, so the reviewers' concerns remain unaddressed.

**Reviewer Scores:**

Since the authors did not submit a rebuttal, it is very unlikely that any of the reviewers will change their score.

---

### Decision · Program_Chairs · 2026-01-26

Reject